# Evaluating the Anticholinergic Burden in Older Patients: Comprehensive Insights from a Nationwide Survey Among Emergency Medicine Specialists in the UK

**DOI:** 10.3390/geriatrics10060137

**Published:** 2025-10-24

**Authors:** Piyawat Dilokthornsakul, Carrie Stewart, Phil Moss, Roy L. Soiza, Fraser Birse, Selvarani Subbarayan, Athagran Nakham, Nantawarn Kitikannakorn, Phyo K. Myint

**Affiliations:** 1Ageing Clinical & Experimental Research Team, Institute of Applied Health Sciences, University of Aberdeen, Aberdeen AB24 3FX, UK; 2Faculty of Pharmacy, Chiang Mai University, Chiang Mai 50200, Thailand; 3St George’s Emergency Department Collaborative Research Group, St George’s Hospital, London SW17 0QT, UK; 4Royal College of Emergency Medicine Trainee Emergency Research Network (TERN), London SE1 1EU, UK; 5North Bristol NHS Trust Emergency Department, Bristol BS10 5NB, UK; 6Faculty of Pharmaceutical Sciences, Naresuan University, Phitsanulok 65000, Thailand

**Keywords:** anticholinergic burden, emergency medicine, older patients, KAP framework, nationwide survey, emergency department

## Abstract

**Introduction**: Older patients are often exposed to medications with anticholinergic activity. Anticholinergic burden (ACB) from medicines is linked to adverse health outcomes. However, healthcare professionals’ familiarity with ACB remains poor, and there is a lack of research on knowledge, attitudes, or practices (KAPs) of ACB among Emergency department (ED) clinicians. **Methods**: A nationwide survey of ACB based on a pilot survey was developed using the KAP framework and assessed for face and content validity by ACB experts. It was distributed to ED physicians across the UK using REDCap through social media and professional networks. **Results**: Among the 108 ED physicians who responded, 54.6% (n = 59) were aware of the term ACB, but 54.2% (n = 32/59) of them were unable to describe it. Their mean scores for quantifying the ACB score and identifying side effects in older patients were 2.9 and 4.1 out of 10, respectively. 88.9% (n = 96) believed that ACB is an important consideration in older patients. 67.6% (n = 73) agreed that awareness of the consequences of prescribing anticholinergic medications to older patients is important. 50% (n = 54) emphasized the importance of being able to assess and quantify the ACB score. Of the 75 physicians who prescribed these medications, 40% (n = 30/75) were unaware of ACB. 38.7% (n = 29/75) rarely considered ACB, 30.7% (n = 23/75) never considered it, and only 1.3% (n = 1/75) always considered it. The majority of respondents (88.9%, n = 96) agreed that more education on ACB is needed in the ED. **Conclusions**: ED physicians in the UK have limited knowledge and awareness of ACB management and prescribing practices for older patients. However, they show positive attitudes towards their role in ACB management and a willingness to receive further education. The low response rate suggests that findings may reflect a motivated subset of clinicians. These results highlight the need for targeted education and further investigation into curricular coverage of prescribing safety and anticholinergic burden.

## 1. Introduction

The UK population at mid-year 2020 was estimated to be 67.1 million. Like many other nations, the age distribution in the UK is shifting towards older demographics. Predictions indicate that by 2050, around 25% of the UK’s population will be 65 years or older, up from approximately 20% in 2019. This shift is due to a combination of reduced fertility rates and increased life expectancy [1]. In 2021–2022, there were 24.4 million attendances at UK Emergency Departments (EDs), and 21.2% of these were by patients aged 65 years and over [2].

Anticholinergics actively compete with acetylcholine to prevent it from binding to cholinergic G protein-coupled muscarinic receptors. These receptors are found in sweat glands, the sympathetic and central nervous systems, and the cell membranes of parasympathetic nervous system cells. Anticholinergic drugs block cholinergic neurotransmission in both the central and peripheral nervous systems, causing various adverse effects [3]. Older patients exhibit increased rates of comorbidity and polypharmacy, making them more likely to be prescribed multiple anticholinergic medications [4]. Approximately 20–50% of older patients are consistently exposed to medications with anticholinergic activity. Additionally, aging individuals may experience a decline in cholinergic function, increased permeability of the blood–brain barrier, and changes in drug pharmacokinetics, all of which can potentially amplify their vulnerability to the effects of anticholinergic medications. Anticholinergic burden (ACB) refers to the cumulative effect of anticholinergic action resulting from the use of multiple medications with varying degrees of anticholinergic activity [5].

ACB scales were developed to quantify the pharmacological effects of anticholinergic medications and to serve as a practical tool for optimizing prescription practices in older patients [6]. The ACB can be quantified using a scoring system that categorises drugs based on their anticholinergic properties: none, possible, or definite. In this system, a score of 0 indicates no anticholinergic properties, 1 indicates possible anticholinergic properties, and scores of 2–3 indicate definite anticholinergic properties [7]. ACB is associated with adverse health outcomes [8] such as cognitive impairment [9], falls [10], delirium [11], hospitalization [12], increased length of hospital stay, greater risk of emergency department visits [13] and increased mortality [14].

However, healthcare professionals’ familiarity with ACB is still poor, and they may overlook the presentation of associated side effects [15]. In addition, there is a notable lack of literature from the ED setting. Given their unique position, Emergency Medicine (EM) physicians may be able to opportunistically perform focused medication reviews considering the ACB when patients present to the ED. This study, via a UK wide survey of EM clinicians, aimed to identify gaps in knowledge, attitudes, or practices (KAPs) with regard to the ACB in older patients. Addressing such gaps, if identified, could help mitigate harm to older patients and reduce subsequent attendances and admissions.

## 2. Materials and Methods

### 2.1. Study Design and Population

This nationwide anonymised online survey received ethical approval from the School Ethics Review Board (SERB) at the School of Medicine, Medical Sciences and Nutrition, University of Aberdeen (Approval number: SERB815161).

#### 2.1.1. Potential Participants

In the UK emergency medicine training pathway, a specialty trainee refers to a doctor enrolled in formal postgraduate training. Registrars are mid- to senior-level trainees, typically in the latter stages of specialty training. Consultants are fully qualified senior physicians who lead clinical teams and supervise trainees.

According to workforce data published by the Royal College of Emergency Medicine in 2018, across the UK the population size of whole-time equivalent specialty trainees in emergency medicine is 1762, and total number of whole-time equivalent consultants in emergency medicine is 2075 [16].

#### 2.1.2. Inclusion Criteria

Specialty trainees or Consultants in Emergency medicine are employed in the NHS.

#### 2.1.3. Exclusion Criteria

Those do not meet the above inclusion criteria.

#### 2.1.4. Recruitment

An invitation email was distributed on the study team’s behalf by the Royal College of Emergency Medicine Trainee Emergency Research Network (TERN). A reminder email was sent after 1 month. The email included an invitation letter, a participant information leaflet (PIL), and the survey URL. The study was also advertised on the X (formerly Twitter) to enhance visibility and reach among emergency medicine professionals. This platform is actively used by the TERN to disseminate research opportunities and engage clinicians across the UK [17]. An invitation letter and the survey URL were posted on X. The participants read the PIL and clicked to give consent on page one of the survey. The survey invitation was open and accessible to anyone via the survey link. Opening questions confirmed eligibility. No incentives were offered for survey completion.

### 2.2. Survey Content and Administration

#### 2.2.1. Survey Content

The survey (Appendix A) was designed using the KAP framework [18]. It was adapted from previous work with the author’s permission (a survey amongst ED staff at St George’s Hospital) [19]. A team of experts in the field of ACB assessed the survey for face and content validity. The question types utilised included tick boxes (true/false and yes/no), multiple-choice, open-ended, Likert scales, and satisfaction ranking (survey feedback). The survey was expected to take around 10 min to complete and included the following sections: (a) Participant demographics (Role & grade of the participant). (b) Participant knowledge of the ACB (6 items). (c) Participant attitude towards the ACB (4 items). (d) Prescribing practices around the ACB (8 items). (e) Survey feedback (2 items). With regard to ACB knowledge we used scoring system available from an online calculator (https://www.acbcalc.com accessed on 29 March 2022). The survey consisted of 9 pages, with each page containing 2–4 items. The survey used a branching structure. If respondents were unaware of the term ‘ACB,’ they were excluded from subsequent knowledge-based questions and redirected to the sections on attitude and clinical practice. Subsequent questions evaluated the proficiency of the ‘aware’ group of physicians in articulating the concept of ‘ACB,’ quantifying the ACB score for 10 commonly administered medications in the ED and assessing their awareness of the potential associated side effects. Participants and the public were not involved in any way in the development of the survey.

#### 2.2.2. Survey Administration

We conducted an anonymised online survey over 3 months (1 May 2024–31 July 2024) targeting specialty trainees and consultants in emergency medicine across the UK. The survey was administered using the REDCap electronic data capture platform hosted at the University of Aberdeen [20,21].

### 2.3. Data Analysis and Reporting

We conducted the data analysis by exporting the raw data from REDCap. We analyzed only completed questionnaires. The quantitative data, primarily categorical, were expressed as percentages with numbers in parentheses. Microsoft Excel was used to generate a combination of pie charts and bar charts to illustrate the quantitative data. (e.g., grammar, spelling, punctuation, and formatting) does not need to be declared.

## 3. Results

### 3.1. Demographics

Among the 108 ED physicians who responded—representing approximately 2.8% of the estimated 3837 eligible emergency medicine consultants and specialty trainees in the UK—the predominant grade was consultants, comprising 50.0% (n = 54), followed by registrars at 34.3% (n = 37) (Table 1).

### 3.2. Knowledge

The initial question inquired whether the physicians were aware of the term ‘ACB’. Of the 108 respondents, 54.6% (n = 59) indicated awareness, while 34.3% (n = 49) did not. The question was, ‘Which best describes your understanding of the ACB?’ Of the 59 physicians who reported being aware of the ACB, 54.2% (n = 32) incorrectly described the term ACB.

The average score of 59 physicians in accurately quantifying the ACB score for ten commonly used medications was 2.9 out of 10 (range 0–7). Several questions asked respondents to correctly identify the ACB scores of medications. Among the ten medications, oxybutynin had the highest ACB score of 3. Of the 59 respondents, 57.6% (n = 34) correctly identified this score, 18.6% (n = 11) provided incorrect scores, and 23.7% (n = 14) were uncertain.

As shown in Figure 1a, for medications with an ACB score of 1 (Table 2), there was an average correct response rate of 15.6% (range 6.8–25.4%). Conversely, an average of 46.1% (range 37.3–61.0%) provided incorrect scores, while 38.3% (range 32.2–44.1%) were uncertain. For medications with an ACB score of 0, there was an average correct response rate of 39.8% (range 20.3–49.2%). Conversely, an average of 21.2% (range 13.6–37.3%) provided incorrect scores, while 39.0% (range 37.3–42.4%) were uncertain. For all medications, the most common response was ‘don’t know,’ with an average of 37.1% (range: 23.7–44.1%). On average, 29.5% (range: 6.8–57.6%) of physicians correctly identified the score, while 33.4% (range 13.6–61.0%) provided incorrect scores.

The average score of 59 physicians in identifying the potential side effects associated with older patients having a high ACB score was 4.1 out of 10 (range 0–8). As shown in Figure 1b, 86.4% (n = 51) correctly identified falls and delirium and 59.3% (n = 35) correctly identified dementia. Cardiovascular disease was correctly identified by 33.9% (n = 20), incorrectly identified by 30.5% (n = 18), and 25.6% (n = 21) were uncertain. Additionally, 16.9% (n = 10) of the physicians correctly identified that stroke is associated with a high ACB score. An average of 49.2% (range 42.4–55.9%) incorrectly identified urinary incontinence, sarcopenia, gastro-oesophageal reflux disease, and open-angle glaucoma as associated with a high ACB score, when they are not known to be. However, 50.8% (n = 30) of physicians correctly indicated that osteoporosis is not associated.

Of the 59 physicians, 70.4% (n = 38) reported that they acquired their knowledge regarding the ACB during postgraduate training.

### 3.3. Attitudes

Among the 108 respondents, 88.9% (n = 96) believed that the ACB is an important consideration in older patients. Additionally, 83.3% (n = 90) of the physicians felt responsible for managing this burden. A total of 67.6% (n = 73) agreed that awareness of the consequences of prescribing anticholinergic medications to older patients is important (Figure 2a). Furthermore, 50% (n = 54) of the physicians emphasized the importance of being able to assess and quantify the ACB score (Figure 2b).

### 3.4. Practice

Among 108 physicians, it was observed that 75 physicians prescribed anticholinergic medications. Among these 75 physicians, 40.0% (n = 30) were part of the 49 physicians who were unaware of the ACB and its potential consequences.

Of those asked to evaluate the extent to which they considered ACB in their clinical practice (n = 75) just over one third (38.7%, n = 29) rarely considered it, 30.7% (n = 23) never considered it, and 24.0% (n = 18) sometimes considered it (Figure 3a). Only 1.3% (n = 1) always considered the ACB in their practice, while 5.3% (n = 4) often considered it.

The frequency with which physicians calculate the ACB score when documenting a patient’s medication history was asked about in the survey. The majority of 81.3% (n = 61) never calculated the ACB score, 14.7% (n = 11) rarely calculated it, and 4.0% (n = 3) sometimes calculated it (Figure 3b). No physicians reported always or often calculating the ACB score in their practice.

Among 75 respondents regarding actions taken after calculating a high ACB score, 61.7% (n = 37) would inform the admitting clinical team, 51.7% (n = 31) would notify the GP, and 21.7% (n = 13) would discuss it with the patient. Additionally, 20.0% (n = 12) do not calculate the ACB score, while 6.7% (n = 4) selected “other” (Figure 3c). Free-text responses indicated that some physicians would also inform the polypharmacy clinic, ED pharmacist, or frailty team.

In the final section of the survey, it was revealed that 88.9% of the 108 physicians (n = 96) agreed that more education is required regarding the ACB in the ED.

## 4. Discussion

Our results indicated a significant knowledge gap regarding the ACB among physicians. The mean score of physicians in accurately quantifying the ACB score for ten commonly used medications and in identifying the potential side effects associated with older patients having a high ACB score was below 5 out of 10. While most physicians could correctly identify the medications with the highest and lowest ACB scores, they often incorrectly identified medications with an ACB score of 1. A large proportion of physicians correctly identified falls, delirium, and dementia as potential side effects. Interestingly, some physicians continued to prescribe anticholinergic medications despite being unaware of the ACB. Even among those who were aware of the ACB (54.6%), these medications were prescribed without utilising anticholinergic scoring.

Although only 54.6% of respondents reported being aware of the term “anticholinergic burden,” 88.9% agreed that ACB is an important consideration in older patients. This apparent discrepancy may reflect a general awareness of medication-related risks in older adults, even if the specific terminology or scoring systems are unfamiliar. It suggests that clinicians recognize the clinical relevance of anticholinergic effects, despite limited formal knowledge of ACB tools.

This study was based on a survey which aimed to assess KAP regarding the ACB among ED staff at St George’s Hospital [19]. This included non-physician clinicians, although physicians constituted the majority at 79.4%. The survey also revealed a substantial knowledge gap concerning the ACB among clinicians. Notably, some staff members, despite being unaware of the ACB, continued to prescribe anticholinergic medications. Even among those aware of the ACB (65.7%), anticholinergic medications were still prescribed without utilising anticholinergic scoring.

These reflect suboptimal clinical practices concerning the ACB, potentially overlooking opportunities to reduce iatrogenic harm in older patients presenting to the ED. There is the potential to explore the use of an ACB tool in the ED. For high-risk patients discharged to their homes, there is potential to create a system that alerts general practitioners about the necessity for a medication review [22]. Most physicians exhibited a positive attitude towards their role in managing the ACB and expressed a strong interest in further education on the topic.

In addition to emergency physicians, hospital pharmacists can play a pivotal role in increasing awareness and application of ACB principles. Their expertise in medication reconciliation and pharmacological risk assessment makes them valuable collaborators in identifying high-risk patients and supporting prescribing decisions [23]. Integration of pharmacists into ED workflows—particularly in frailty or polypharmacy teams—could enhance ACB screening and facilitate safer transitions of care.

Given the fact that older patients are major users of emergency services, identifying patients with high ACB at ED attendance may land itself as a great opportunity to red flag the community team, such as general practitioners and community pharmacists, for medicine optimisation as appropriate. This is of particular import as many side effects of ACB present with common problems such as delirium triggered by constipation and urinary retention and falls which may have potential to prevent further falls with appropriate medicine optimisation.

We reviewed the RCEM curriculum [24] and found that while prescribing safety and pharmacology are included as core competencies, anticholinergic burden is not explicitly addressed. The curriculum does not specify the number of hours or formats (e.g., lectures, tutorials, bedside teaching) dedicated to these topics. This omission may contribute to the observed knowledge gaps. Future studies should explore how emergency medicine trainees receive and evaluate this content—ideally through structured interviews or focus groups with registrars nearing completion of training.

A limitation of this study is that, methodologically, the survey instrument had limited formal validation. Due to our focus on specialty trainees, we excluded many non-training-grade doctors working in UK EDs. This exclusion may impact the number of participants and may not accurately reflect the true context of UK EDs. However, based on previous local survey (all ED clinicians) and the current study findings, it is reasonable to conclude that the findings are likely to reflect current knowledge, attitude and practice of all physicians (and clinicians) working in the UK. Additionally, the small sample size precluded meaningful comparisons between different grades of physicians due to their disproportionate representation. More respondents would be needed to meaningfully compare the staff groups. Our nationwide survey achieved 108 respondents, yielding a response rate of 2.8%, which aligns with other surveys distributed via the Trauma Audit and Research Network (TARN) and the Trainee Emergency Research Network (TERN). For comparison, Battle et al. [17]. reported 113 UK respondents with broad geographical coverage, while Foley et al. [25] recorded 66 UK respondents within a larger international sample. In both studies, the total number of clinicians who received the survey was not identifiable, preventing accurate response rate calculation and underscoring a common limitation of surveys disseminated through professional network. As with our study, the low participation likely reflects a self-selected group of motivated clinicians, which may limit the generalizability of findings.

## 5. Conclusions

Emergency physicians in the UK demonstrate limited knowledge and prescribing awareness regarding anticholinergic burden, despite expressing positive attitudes and a willingness to learn. The low survey response rate underscores the need for broader engagement and suggests that current findings may reflect a motivated minority. Our review of the RCEM curriculum indicates that ACB is not explicitly covered, pointing to a potential gap in formal training. Future research should include qualitative interviews with registrar trainees to assess how prescribing safety is taught and perceived. Addressing these gaps through targeted education and system-level interventions may improve outcomes for older patients and reduce avoidable ED visits and admissions.

## Figures and Tables

**Figure 1 geriatrics-10-00137-f001:**
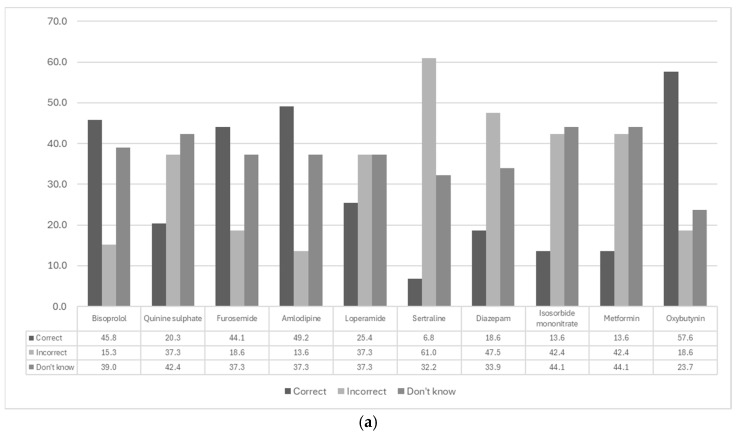
Analysis of ED physicians’ responses to knowledge-based survey questions. (**a**) Bar chart illustrates the accuracy of physicians in quantifying the ACB score for ten commonly used medications, categorised into four scoring groups (Table 2). (**b**) Evaluation of physicians’ ability to identify conditions associated with high ACB scores in older patients. The conditions listed in the bottom five rows are associated with high ACB scores, whereas those in the top five rows are not (n = 59). CVD = Cardiovascular disease, GERD = Gastro-esophageal reflux disease.

**Figure 2 geriatrics-10-00137-f002:**
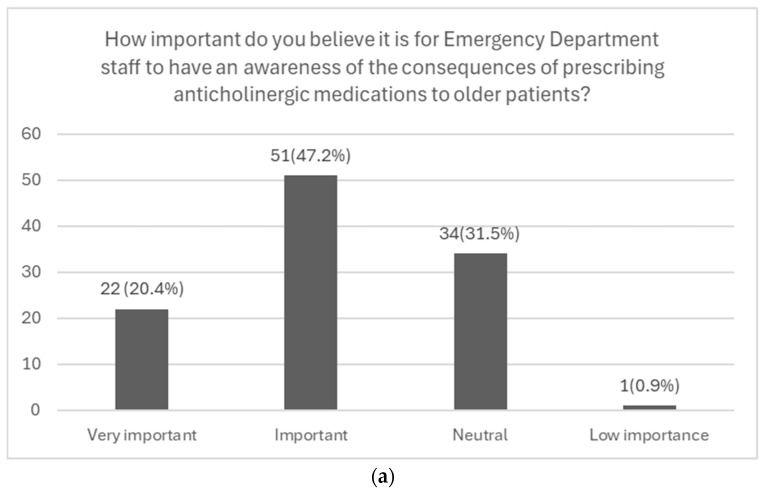
Graphical representations of emergency department physicians’ responses to questions from the ‘Attitudes’ section of the survey. (**a**) Likert scale illustrating physicians’ awareness of the potential consequences of their prescribing practices. (**b**) Likert scale illustrating physicians’ awareness of their ability to assess and quantify the ACB in patients (n = 108).

**Figure 3 geriatrics-10-00137-f003:**
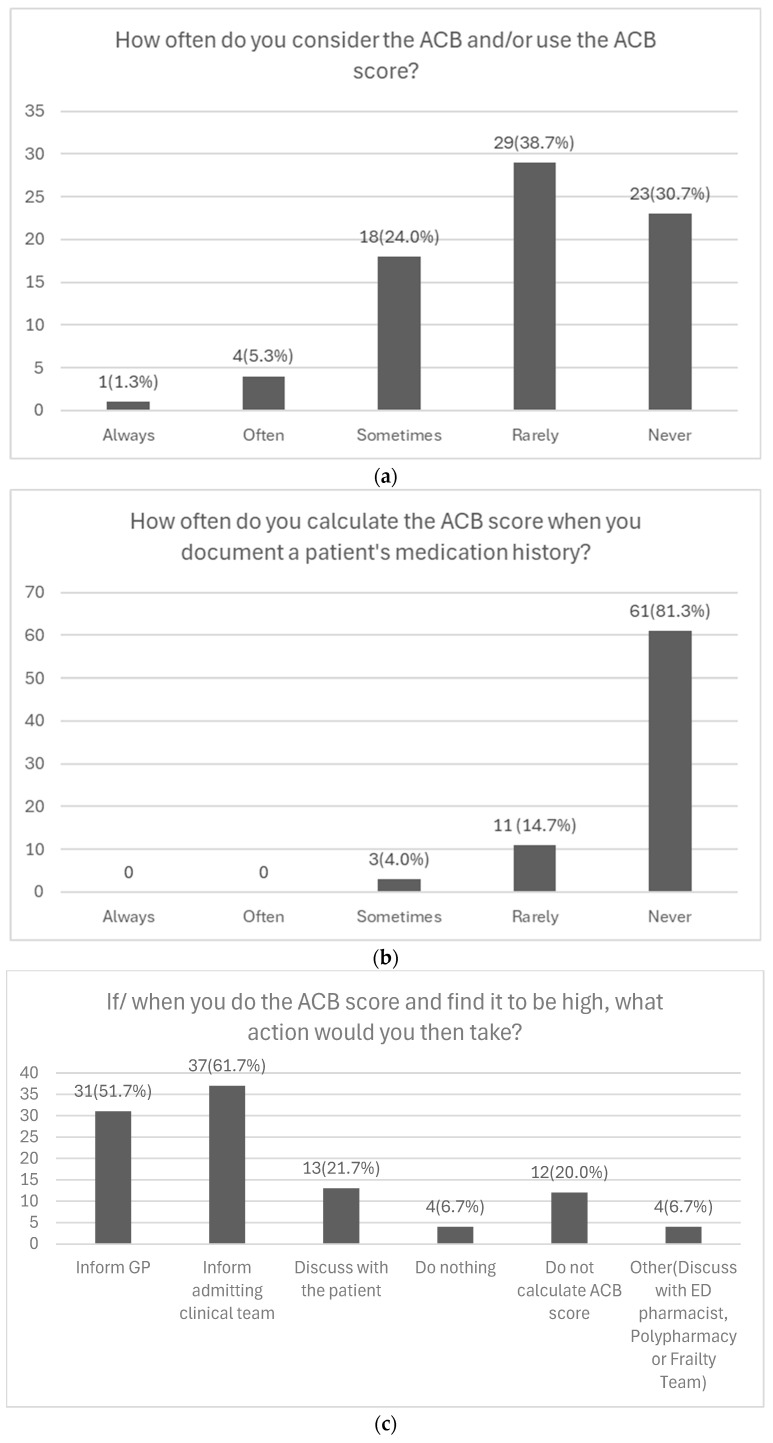
Analysis of Physicians’ Responses to Questions from the ‘Clinical Practice’ Section of the Survey. (**a**) Depiction of the frequency with which physicians consider the ACB in their prescribing practices (n = 75). (**b**) Depiction of the frequency with which physicians calculate the ACB score when documenting a patient’s medication history (n = 75). (**c**) Depiction of the actions physicians would take upon identifying a high ACB score in their patients (n = 60).

**Table 1 geriatrics-10-00137-t001:** Demographics.

Grade of Participants	n (%)
Consultant	54 (50.0)
Core trainee	17 (15.7)
Registrar	37 (34.3)
Total	108 (100)

**Table 2 geriatrics-10-00137-t002:** Anticholinergic burden (ACB) score of the 10 medications listed in the survey.

ACB Score *	0	1	2	3
Medication	Bisoprolol	Metformin		Oxybutynin
	Quinine sulphate	Diazepam		
	Furosemide	Isosorbide mononitrate		
	Amlodipine	Loperamide		

* ACB scores of the 10 medications are based on the ACB calculator [7].

## Data Availability

The raw data supporting the conclusions of this article will be made available by the authors on request.

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
