# Peer review of "Evaluating the Anticholinergic Burden in Older Patients: Comprehensive Insights from a Nationwide Survey Among Emergency Medicine Specialists in the UK"

_geriatrics, 2025, doi:10.3390/geriatrics10060137_

Round 1

Reviewer 1 Report

Comments and Suggestions for Authors

Well done! You have unearthed a problem of great importance. It is shocking that of 1762 trainees and 2075 consultants (= 3,837) only 108 = 2.8% responded. You must emphasise how limited your sample is, and probably represents the motivated respondents.

  1. Can you find the response rate of other surveys sent out by the the Royal College and the Emergency group of the College?
  2. Can you obtain the curriculum for the emergency registrars?
  3. Using that curriculum can you interview a random sample of registrars especially those near completion asking (a) how much of the curriculum did they receive and was it in formal lectures, tutorials... or on the job? (b) how do they assess the quality of what they received. See if you can find a formal assessment tool so the rankings could be compared to other surveys.
  4. For this article stress the tiny response and the low knowledge scores. See if you can obtain the curriculum and find out how many hours and in what format are allocated to pharmacology/prescribing. 
  5. For your next study interview the registrar trainees!!

I have been the head of a University Family Medicine department, Head of the Residency programme and Head of the research Department in different hospitals so the approach I would take is find the documentation and find the registrar response. We found if you got all the registrars together in less than an hour you could find all the strengths and weaknesses of the instructional and clinical programmes. Needs diplomacy!

Reviewer 2 Report

Comments and Suggestions for Authors

ACB is an important problem that should be taken into account, especially in older people. However, in general is not usually considerated. Therefore, I think that this survey can make in value the importance of having knowledge about ACB and its consideration when prescribing.

The study is well performed, however there are some point that need clarification or improvement.

Material and Methods:

  • The authors should explain the meaning of seciality tainees, cosultant, registar as these are positions or levels of training in the UK but not in other countries.
  • Why did the authors though necessary to advertise the survey in X?

Results:

  • Lines 145-147 should be rewritten as it is not clear the meaning
  • I think there is somme contradiction in the fact that only 54,6% of the respondents are aware of ACB but 88,9% consider that ACB is important. The authors should give, if possible, some explanation about this
  • Figure 1 data are difficult to read. 

Discussion

Do the authors have though that hospital pharmacists can be of help in increasing ACB awarrness and its aplication? 

Round 2

Reviewer 1 Report

Comments and Suggestions for Authors

Thanks to the authors for their emendations

Reviewer 2 Report

Comments and Suggestions for Authors

The authors have answered all que questions that I made in my review in a positive way and have made substantial changes in the manuscript.